# Image Servo Tracking of a Flexible Manipulator Prototype with Connected Continuum Kinematic Modules

Ming-Hong Hsu [1], Phuc Thanh-Thien Nguyen [1], Dai-Dong Nguyen [1] and Chung-Hsien Kuo [2,*]

1   Department of Electrical Engineering, National Taiwan University of Science and Technology, Taipei 106, Taiwan
2   Department of Mechanical Engineering, National Taiwan University, Taipei 106, Taiwan
*   Correspondence: chunghsien@ntu.edu.tw

**Abstract:** This paper presents the design and implementation of a flexible manipulator formed of connected continuum kinematic modules (CKMs) to ease the fabrication of a continuum robot with multiple degrees of freedom. The CKM consists of five sequentially arranged circular plates, four universal joints intermediately connecting five circular plates, three individual actuated tension cables, and compression springs surrounding the tension cables. The base and movable circular plates are used to connect the robot platform or the neighboring CKM. All tension cables are controlled via linear actuators at a distal site. To demonstrate the function and feasibility of the proposed CKM, the kinematics of the continuum manipulator were verified through a kinematic simulation at different end velocities. The correctness of the manipulator posture was confirmed through the kinematic simulation. Then, a continuum robot formed with three CKMs is fabricated to perform Jacobian-based image servo tracking tasks. For the eye-to-hand (ETH) experiment, a heart shape trajectory was tracked to verify the precision of the kinematics, which achieved an endpoint error of 4.03 in Root Mean Square Error (RMSE). For the eye-in-hand (EIH) plugging-in/unplugging experiment, the accuracy of the image servo tracking system was demonstrated in extensive tolerance conditions, with processing times as low as $58 \pm 2.12$ s and $83 \pm 6.87$ s at the 90% confidence level in unplugging and plugging-in tasks, respectively. Finally, quantitative tracking error analyses are provided to evaluate the overall performance.

**Keywords:** continuum robot; image servo tracking; image jacobian; autonomous manipulation

## 1. Introduction

### 1.1. Motivation

Recently, automation-related equipment, especially robotic manipulators, has been successfully used widely in industrial applications [1,2], medical services [3], and disaster-based narrow-space exploration [4]. In addition to the conventional articulated manipulators, soft and flexible manipulators are becoming more attractive to robotics researchers [5,6]. Continuum robots are typical examples of flexible manipulators, which have better agility than conventional articulated manipulators. In general, a continuum robot is practical for performing soft grasping and manipulating tasks, in the same vein as the manipulations of the octopus tentacle or the elephant trunk. Hence, continuum robots are often operated in narrow spaces because of their hyper degrees of freedom (DOF) motions [7,8]. Although providing flexibility and agility in manipulating tasks, the deployment of multi-segment continuum manipulators that exhibit non-constant curvature is still a formidable challenge because of the complexity of deriving continuum kinematics.

### 1.2. Related Works

Continuum robots are typical examples of flexible manipulators capable of performing with better agility than conventional articulated manipulators. The designs of continuum

robots can be divided into two types: tension cable-driven and pneumatic-driven. For tension cable-driven robots, Hannan et al. [9] designed a continuum robot formed by 32 coupled joints and a total of eight actuatable cables, making it capable of performing a motion similar to that of an elephant trunk. Jones et al. [10] further proposed synthesizing the kinematic relationship between a general continuum skeleton and a continuum robot. The coordinates are related to the input of the controller (e.g., air pressure and tendon length) to realize real-time tasks and shape control of a continuum robot. As Yoon et al. [11] proposed, springs are also used as the backbone of the plate connection. Such a mechanism design is mainly used to enhance the elasticity and flexibility of the overall continuum structure to ensure the safety of the continuum robot when it collides with the human body. Another design proposed by Tokunaga et al. [12] used an elastic center column and springs, where the springs are installed outside the center column to avoid unexpected shapes such as twisting, which is beneficial for handling weights and loads on the end effector. For pneumatic-driven robots, the use of pneumatic artificial muscles is available for developing continuum robots. Liu et al. [13] presented a light soft manipulator, where thin McKibben pneumatic artificial muscles were utilized for continuously controllable stiffness actuation. Their study successfully overcomes problems such as that most continuum robots cannot continuously change their stiffness at a fixed end position. Dalvand et al. [14] proposed an analytical loading model by considering the number of tendons and the load distribution of the tendons to avoid the tendon relaxation problem that leads to inaccurate motion control. Pneumatic-type continuum robots need a pneumatic source, and, therefore, they are not convenient for installation, mobility, and maintenance. However, the tension cable-driven type usually uses motors or linear actuators as the source of kinematic motion control. Consequently, tension cable-driven continuum robots have advantages in terms of installation, mobility, and maintenance. However, the kinematics of the tension cable-driven continuum robot is much more complicated than those of conventional articulated robots.

In addition, bioinspired continuum robot designs have been studied. For example, Hassan et al. [15] developed an active-braid design to fabricate a bioinspired continuum manipulator. Their study adopted flexible cross-linked spiral array structures to form the continuum structure. Inspired by the biological structure of snakes, Zhang et al. [16] proposed a compound continuum robot combining concentric tubes and a notched continuum robot to achieve a smaller diameter and a larger central cavity.

Because continuum robots can navigate and manipulate in a narrow space, they have attracted considerable attention in designing surgical robots. Safety is an essential aspect of robotic surgery. Comin et al. [17] presented a solution of combining a pneumatic soft continuum robot and a rigid robot arm in terms of series connection. The rigid part maintains a safe tool contact force, while the soft part follows the required cutting path. Their solution can demonstrate teleoperated diathermic tissue-cutting tasks from a safety consideration; although, the proposed system still was not friendly-using and only worked on designed scenarios. In addition, Zhao et al. [18] proposed a variable stiffness design for a continuum manipulator. Such a redundant continuum structure is formed with an elastic backbone that exhibits a continuously constrained bending curvature. Agility and miniaturization are design considerations in minimally invasive surgery (MIS) operations. Qu et al. [19] presented a continuum manipulator design for MIS operation, with a bioinspired wire-driven multi-backbone structure design being applied in this study. A super-elastic backbone was utilized to connect a series of thin disks to form a continuum structure with a 12 mm outer diameter and a total length of 180 mm. Another study was proposed for flexible endoscopic surgery purposes. Hwang et al. [20] designed a novel constrained continuum manipulator that used several auxiliary links attached to the primary continuum for payload capability improvement. Similar to the design of three 6-DOF continuum robots formed with cable-driven concentric tube mechanisms [21], the design of a snake-like surgical continuum robot [22] and a continuum manipulator with parallel and shifted-routing cable-driven control mechanisms for robotic surgeries of endometrial regeneration [23], provide the design paradigms for novel continuum

manipulators. Although making some achievements, these methods had to consider finding a balance between the manipulator's workspace and stiffness without increasing the system's complexity.

Furthermore, to survey the structural designs of continuum manipulators, the control designs of continuum manipulators are also discussed. Shen et al. [24] presented a study on a flexible backbone cable-driven continuum manipulator to improve accuracy to overcome such factors as gravity or mechanism effects. The authors proposed a method consisting of a kinematic model and data-driven Gaussian process regression (GPR) based on experiments applied to hardware platforms to reduce the position and orientation errors by 68.72% and 51.74%, respectively. Other control systems applied to continuum robots also provide helpful information in designing continuum robot control systems, such as pose planning of a multi-section continuum manipulator in terms of an imitation learning-based approach [25] and absolute positioning accuracy improvement of a continuum surgical manipulator by utilizing the closed-loop control approach [26].

Images-based visual servoing (IBVS) studies are also discussed to demonstrate path tracking and autonomous manipulation abilities. Lai et al. [27] proposed a vision-based adaptive control scheme based on a soft continuum manipulator with a bidirectional two-segment configuration. The proposed controller was realized with the IBVS approach. The key point positioning error could reach within 6.5% based on the manipulator length, and the robot will find the best fit to the desired shape if the goal positions are not reachable. Yang et al. [28] presented the stereo tracking of a continuum surgical manipulator. A wrist marker was designed to realize the closed-loop image-based servoing control scheme. The feature points extracted from the stereoscopic images were evaluated to align with the actual pose, including the position and direction of the target. Although the tip positioning error was reduced to 25.23% during trajectory tracking, the control cycle is longer than the preferred, making the manipulator overshoot when it changed motion direction. Finally, many studies were proposed to realize the visual servoing ability of continuum robots for a wide variety of applications, such as visual servoing and compliant control [29], image-based laser beam steering control [30], model-free visual servoing combined with singularity avoidance to enhance safety [31], and optical coherence tomography (OCT)-guided visual servoing for micromotion manipulations [32,33], vision-based shape control for cable-driven manipulator [34], hybrid EIH and ETH for visual servoing [35], and OCT path scanning [36].

### 1.3. Contribution

This paper proposes an ideal constant curvature assumption to overcome the problem of deriving continuum kinematics. With the well-defined continuum kinematics in [9] and [10], this paper presents the kinematics of a continuum kinematic module (CKM) via the parameters $\kappa$, $\varphi$, and $l$ (the arc length, curvature, and rotation angle in the x-y plane of CKM, respectively). Therefore, a complete continuum robot formed of connected CKMs can be accumulated. It is noted that three CKMs are utilized to produce a continuum robot for image-based servo tracking practices. Moreover, image-based path tracking and autonomous manipulation are typical demonstrations for investigating the applicability of hyper DOF continuum manipulators for possible deployment of service robots to provide manipulation compliance and ensure safety in operation when performing human-robot interaction (HRI). The main contribution of this study is summarized as follows:

- This paper presents a three-segment CKM-based continuum robot design that is convenient for the fabrication of a continuum robot and efficient at obtaining the overall kinematic model for control purposes.
- For the proposed design and control validation, experiments of image-based servoing path tracking and autonomous manipulation are established, utilizing Jacobian images to track the desired image targets by controlling the continuum robot. In the eye-to-hand (ETH) experiment, a heart shape trajectory was tracked to verify the precision of the kinematics with acceptably low endpoint errors. In the eye-in-hand (EIH)

experiment, a stereo vision-based object detection algorithm was developed for the power socket grasping task with high accuracy and efficient operation in real-time.

### 1.4. Limitation

Although addressing the problem of deriving continuum kinematics, the wire-driven CKM causes posture deviation of the proposed three-segment continuum robot due to the influence of gravity. Therefore, the mentioned limit may lead to the performance of the visual servoing system in the EIH plugging/unplugging experiment being imperfect under low-tolerance conditions.

The remainder of this study is organized as follows: In Section 2, the detail of the multi-segment CKM-based continuum robot design, the architecture of a single CKM, and the implementation of the image-based servo tracking systems are carefully described. Next, Section 3 presents the experimental materials and methods, followed by the results and corresponding subsequent analyses. Finally, the conclusions and future work are summarized in Section 4.

### 2. Proposed Method

In this study, the proposed flexible manipulator with visual servoing, as shown in Figure 1, comprises three continuum kinematic modules (CKMs). Each CKM includes five circular plates, four universal joints connecting the circular plate, three independent tension cables, and compression springs surrounding the tension cables. The architecture of a single CKM is extended to the proposed flexible manipulator (i.e., continuum robot) composed of three CKMs.

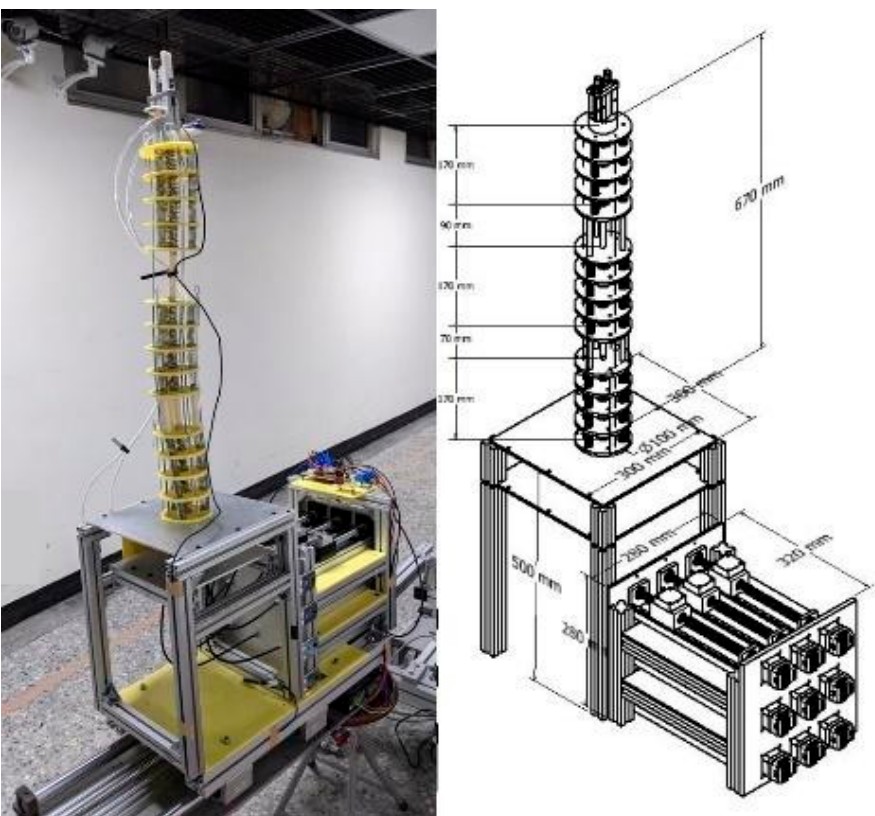

**Figure 1.** The proposed flexible manipulator with three connected CKMs.

The visual servo control element deals with image-related information, identification and selection of tracking points, Image Jacobian, and velocity kinematics calculations. For the visual servo control to be able to achieve real-time processing effects, this system uses a

Win10 computer, based around an i7-9750H CPU and NVIDIA GTX1660 GPU, and an Intel Realsense D435i camera.

The continuum-manipulator platform is controlled based on the microcontroller Teensy 4.0, which is mainly responsible for calculating forward and inverse kinematics. The control of the sliding table motor and air pressure-related components, the signal processing of the sensor, and sending of the control signal to each sliding table, allow the obtaining of the status of each sliding table. Finally, serial communication is used as the communication protocol between the two systems. Figure 2 presents the overall structure of the whole proposed system.

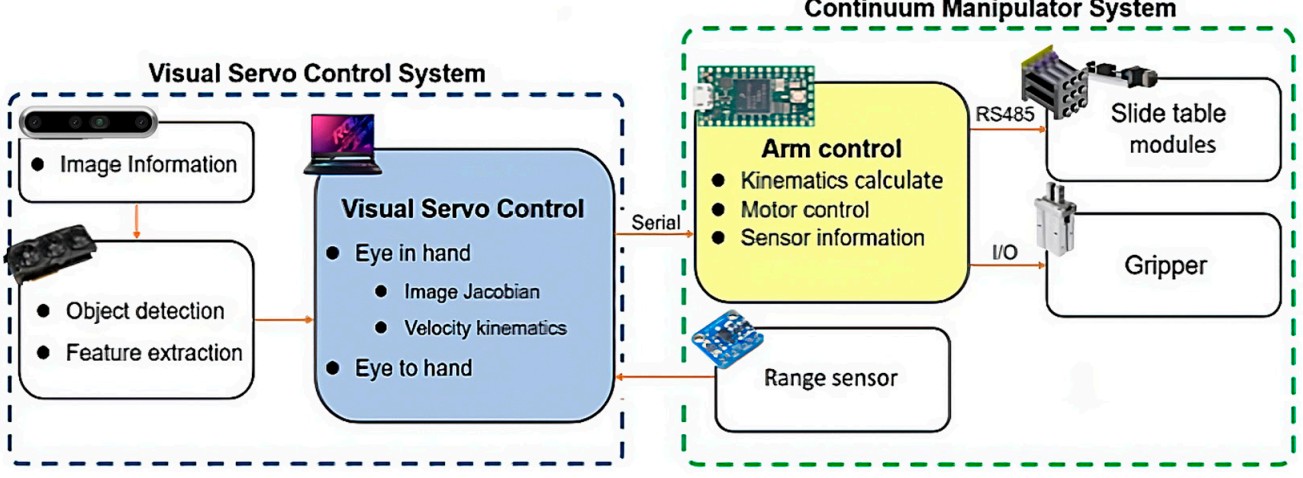

**Figure 2.** The overall architecture of the proposed visual servo continuum manipulator.

The detailed CKM architecture, the design and kinematics of the continuum robot, and image servo tracking will be introduced in the following subsections.

## 2.1. Design and Kinematics of A Continuum Kinematic Module (CKM)

Each CKM is composed of five circular plates and three tension cables, as shown in Figure 3. The backbone of the CKM is formed with universal joints and compressive springs. Tension cables are then used to control the curvature of the CKM. The compressive springs and tension cables are intermediately arranged at the outer ring of the circular plates to stabilize CKM backbone deformation. Three tension cables are then symmetrically located at L1, L2, and L3, and three compressive springs are symmetrically located at L4, L5, and L6.

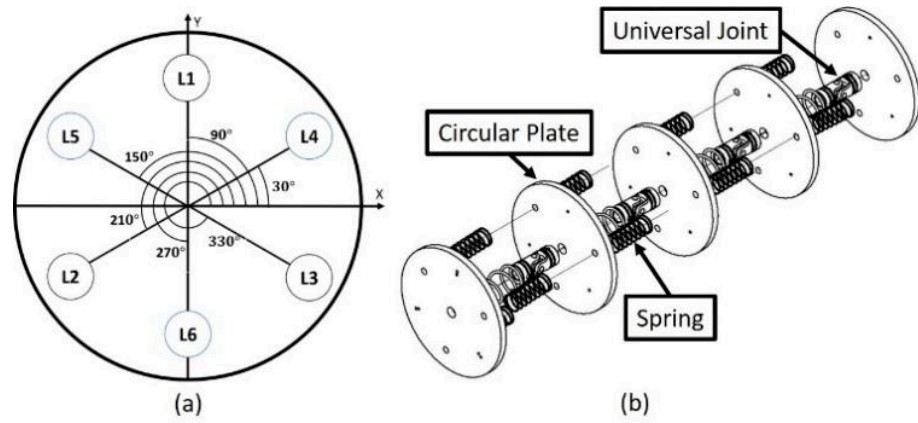

**Figure 3.** (**a**) Position of the compression springs and tension cables; (**b**) composition of a CKM.

The forward kinematics of the proposed CKM is obtained by referring to [9]. First, assume that the CKM bends as an ideal curvature shape. Then, for the convenience of calculation, assume the arc centerline length is $l$, the arc radius is $r$, the angle of the arc is $\theta$, the arc curvature is $\kappa$, the center of the arc is $o$, and the angle that the CKM rotates on the x-y plane is $\varphi$. The illustration of the arc parameter $(\varphi, \kappa, l)$ is shown in Figure 4.

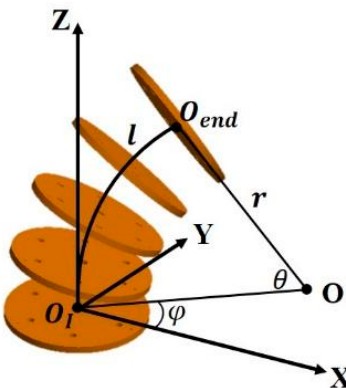

**Figure 4.** Illustration of the CKM kinematics parameters.

As in [37], the rotation of the arm in space is divided into five groups of Denavit–Hartenburg parameters. As illustrated in Figure 4, in the first turn, $\theta$ rotates to an angle of $\varphi$ along the z-axis and $\alpha$ rotates to an angle of $\pi/2$ along the x-axis in space. In the second turn, $\theta$ is rotated to an angle of $\kappa l/2$ along the z-axis while $\alpha$ is rotated to an angle of $-\pi/2$ along the x-axis, which means that the z-axis is aligned with the endpoint. In the third turn, the linear distance between the bottom and the end $d_3$ is extended along the z-axis, and $\alpha$ is rotated at an angle of $\pi/2$ along the x-axis. In the fourth turn, $\theta$ is rotated along the z-axis and $\alpha$ is rotated at an angle of $-\pi/2$ along the x-axis so that the coordinate axis is positioned in the positive direction toward the endpoint of the CKM. Finally, $\theta$ is rotated at an angle of $-\varphi$ to offset the first set of rotations in the last turn. In summary, the D–H matrix of a single CKM can be obtained through the abovementioned rotation relationship, as in Table 1.

**Table 1.** Single CKM D–H matrix.

| Transform Turn | $\theta$ | $d$ | $a$ | $\alpha$ |
|---|---|---|---|---|
| 1 | $\varphi$ | 0 | 0 | $\pi/2$ |
| 2 | $\kappa l/2$ | 0 | 0 | $-\pi/2$ |
| 3 | 0 | $2/\kappa \times \sin(\kappa l/2)$ | 0 | $\pi/2$ |
| 4 | $\kappa l/2$ | 0 | 0 | $-\pi/2$ |
| 5 | $-\varphi$ | 0 | 0 | 0 |

Through the D–H matrix in Table 1, and by substituting it into the calculation, the transformation matrix from the center coordinates of the circular bottom plate to the center coordinates of the end circular plate can be obtained:

$$T_1^5 = \begin{bmatrix} \cos^2\Phi(\cos(kl)-1)+1 & \sin\Phi\cos\Phi(\cos(kl)-1) & \cos\Phi\sin(kl) & \frac{\cos\Phi(1-\sin(kl))}{k} \\ \sin\Phi\cos\Phi(\cos(kl)-1) & \cos^2\Phi(1-\cos(kl))+\cos(kl) & \sin\Phi\sin(kl) & \frac{\sin\Phi(1-\cos(kl))}{k} \\ -\cos\Phi\sin(kl) & -\sin\Phi\sin(kl) & \cos(kl) & \frac{\sin(kl)}{k} \\ 0 & 0 & 0 & 1 \end{bmatrix} \tag{1}$$

The points $(x, y, z)$ in the working plane must be converted into arc parameters $(\varphi, \kappa, l)$ with inverse kinematics using the inverse kinematics in [2]. The conversion shows

its bending geometric relationship, as illustrated in Figure 5. The target point can be projected from three-dimensional space to two-dimensional space:

$$\varphi = \tan^{-1}\frac{y}{x} \tag{2}$$

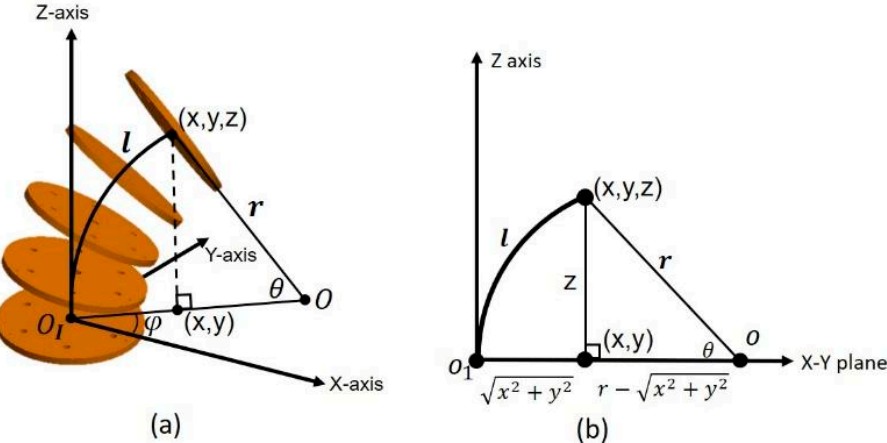

**Figure 5.** The posture of a single CKM reaching the target point: (**a**) three-dimensional and (**b**) two-dimensional.

A single CKM bends in an ideal arc shape, so the intersection of the end circular plate and the bottom circular plate extension line is the arc center of the CKM centerline. Therefore, the connecting line from the circular bottom plate to the center of arc $o$ passes through the projected point to the x-y plane $(x, y)$. Through geometric relations, the arc angle $\theta$ and arc radius $r$ can be obtained:

$$r = \frac{x^2 + y^2 + z^2}{2\sqrt{x^2 + y^2}} \tag{3}$$

$$\theta = \cos^{-1}\left(\frac{r - \sqrt{x^2 + y^2}}{r}\right) \tag{4}$$

The arc curvature $\kappa$ is the reciprocal of the arc radius $r$. Then, the arc length $l$ can be expressed as:

$$l = r\theta = \frac{\cos^{-1}(1 - \kappa\sqrt{x^2 + y^2})}{\kappa}; \ \kappa = \frac{1}{r} \tag{5}$$

The arc parameters $(\varphi, \kappa, l)$ are used in kinematics. However, linear actuators mainly control tension cables to manipulate the CKM. Therefore, the arc parameters $(\varphi, \kappa, l)$ need to be converted into the length of each tension cable, which controls the CKM. The position of the tension cable is the three vertices of the equilateral triangle, so the arc length $l$ is the average length of the three control cables (i.e., $l_{C1}$, $l_{C2}$, $l_{C3}$), as.

$$l = \frac{l_{C1} + l_{C2} + l_{C3}}{3} \tag{6}$$

Figure 6a explains the relation between the tension cables and the arc parameters in space. From Figure 6b, it is determined that $\varphi_{C1} = 90° - \varphi$, and $\varphi_{C2}$ and $\varphi_{C3}$ can be deduced through a geometric relation:

$$\begin{cases} \varphi_{C2} = 210° - \varphi \\ \varphi_{C3} = 330° - \varphi \end{cases} \tag{7}$$

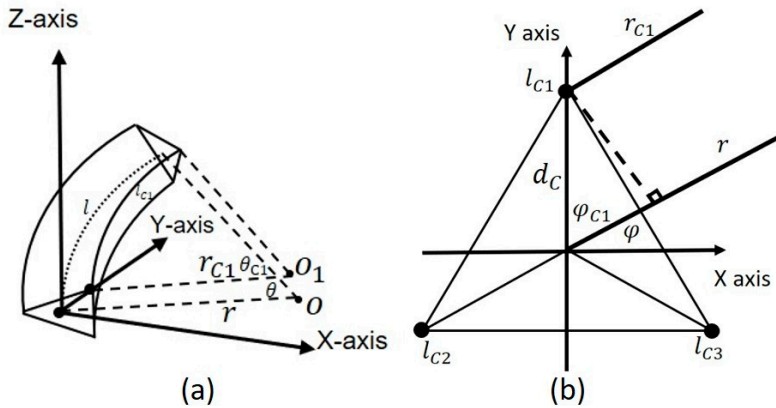

**Figure 6.** Relation between the tension cables and arc parameters: (**a**) three-dimensional and (**b**) two-dimensional.

Then, the arc radius of the tension cable $r_{C1}$ and the central arc radius $r$ are two parallel lines; the relation between $r_{C1}$ and $r$ is expressed through the trigonometric function:

$$r = r_{C1} + d_C * \cos \varphi_{C1} \tag{8}$$

where $d_C$ is the distance between each cable and the center of the circular plate. Then, each arc radius of the tension cables $r_{Ci}$ can be derived from the relation with the central arc radius $r$:

$$l_{Ci} = l - \theta d_C \cos \varphi_i; i = \{1, 2, 3\} \tag{9}$$

The arc length of each tension cable $l_{Ci}$ can be deduced as:

$$r_{Ci} = r - d_C \cos \varphi_i; \; i = \{1, 2, 3\} \tag{10}$$

*2.2. Design and Kinematics of the Flexible Manipulator Formed with Three CKMs*

In this paper, the flexible manipulator consists of three CKMs. As Figure 7a shows, the tendon cables of each segment are controlled via linear actuators. The first circular plate segment passes through nine tendon cables, the second through six segments, and the third through three. Therefore, the design of each CKM circular plate is slightly different, as shown in Figure 7b.

The forward kinematics of the continuum robot is calculated. According to the mechanism configuration, the relationship of each coordinate system is shown in Figure 8.

The coordinate system of the first and third CKMs is the same, while the coordinate system of the second CKM is 180° inversed from the others. Because each segment is calculated independently, the CKM transformation matrix is obtained from the forward kinematic Equation (1). Therefore, each segment and fixed-segment transformation matrix, $T_{si}$ and $T_{di}$ are stated as the following:

$$T_{si} = \begin{bmatrix} \cos^2 \varphi_i (\cos \kappa_i l_i - 1) + 1 & \sin \varphi_i \cos \varphi_i (\cos \kappa_i l_i - 1) & \cos \varphi_i \sin \kappa_i l_i & \frac{\cos \varphi_i (1 - \cos \kappa_i l_i)}{\kappa_i} \\ \sin \varphi_i \cos \varphi_i (\cos \kappa_i l_i - 1) & \cos^2 \varphi_i (1 - \cos \kappa_i l_i) + \cos \kappa_i l_i & \sin \varphi_i \sin \kappa_i l_i & \frac{\sin \varphi_i (1 - \cos \kappa_i l_i)}{\kappa_i} \\ -\cos \varphi_i \sin \kappa_i l_i & -\sin \varphi_i \sin \kappa_i l_i & \cos \kappa_i l_i & \frac{\sin \kappa_i l_i}{\kappa_i} \\ 0 & 0 & 0 & 1 \end{bmatrix} \tag{11}$$

$$T_{di} = \begin{bmatrix} -1 & 0 & 0 & 0 \\ 0 & -1 & 0 & 0 \\ 0 & 0 & 1 & d_i \\ 0 & 0 & 0 & 1 \end{bmatrix} \tag{12}$$

Finally, the transformation matrix of the continuum robot can be obtained by multiplying $T_{si}$ and $T_{di}$:

$$T_m = T_{s_1} T_{d_1} T_{s_2} T_{d_2} T_{s_3} = \begin{bmatrix} a_{11} & a_{12} & a_{13} & a_{14} \\ a_{21} & a_{22} & a_{23} & a_{24} \\ a_{31} & a_{32} & a_{33} & a_{34} \\ a_{41} & a_{42} & a_{43} & a_{44} \end{bmatrix} \tag{13}$$

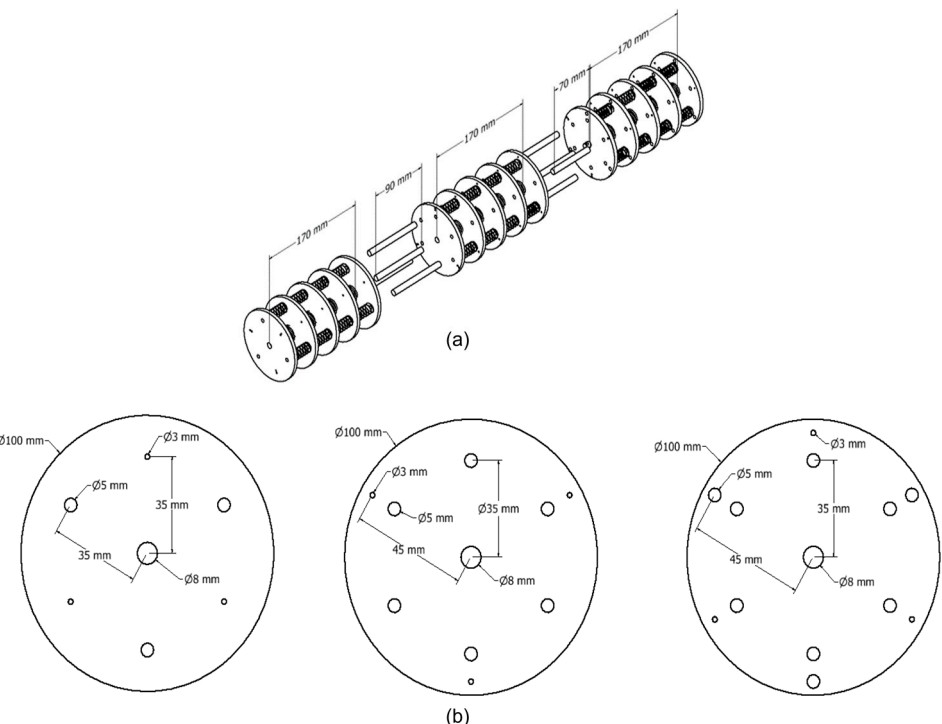

**Figure 7.** Relation between the tension cables and arc parameters: (**a**) three-dimensional and (**b**) two-dimensional.

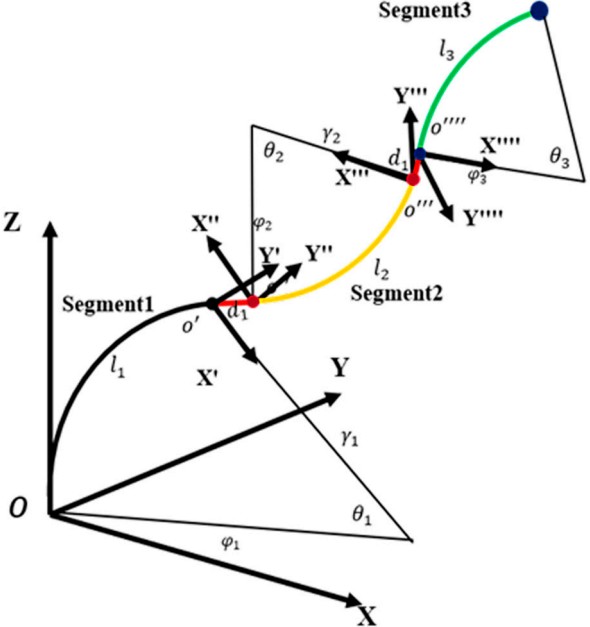

**Figure 8.** The coordinate system of the three-segment continuum robot.

According to [1], the velocity kinematics $v_{jacobian}$ can be written as Equation (14), with $x$ as a vector in the task space, including the position or both the position and direction; $\dot{x}$ implies the $x$ differential with time or velocity, $\dot{q}$ is the change of the arm arc parameters of each axis as Equation (15), and J is the Jacobian matrix:

$$v_{jacobian} = \begin{bmatrix} v_x & v_y & v_z & \omega_x & \omega_y & \omega_z \end{bmatrix}^T = \dot{x} = J\dot{q} \tag{14}$$

$$q = \begin{bmatrix} \varphi_1 & k_1 & l_1 & \varphi_2 & k_2 & l_2 & \varphi_3 & k_3 & l_3 \end{bmatrix} \tag{15}$$

Finally, the transformation matrix of the continuum robot can be differentiated by the chain rule:

$$\dot{T}_{s_1}^{s_3} = \dot{T}_{s_1}T_{d_1}^{s_3} + T_{s_1}\dot{T}_{d_1}T_{s_2}^{s_3} + T_{s_1}^{d_1}\dot{T}_{s_2}T_{d_2}^{s_3} + T_{s_1}^{s_2}\dot{T}_{d_2}T_{s_3} + T_{s_1}^{d_2}\dot{T}_{s_3} = \begin{bmatrix} \alpha_{11} & \alpha_{12} & \alpha_{13} & \alpha_{14} \\ \alpha_{21} & \alpha_{22} & \alpha_{23} & \alpha_{24} \\ \alpha_{31} & \alpha_{32} & \alpha_{33} & \alpha_{34} \\ \alpha_{41} & \alpha_{42} & \alpha_{43} & \alpha_{44} \end{bmatrix} \tag{16}$$

$$J\dot{q} = \begin{bmatrix} \alpha_{14} & \alpha_{24} & \alpha_{34} \end{bmatrix}^T \tag{17}$$

As shown in (14), $v_{jacobian}$ includes a linear and angular velocity, so $x = \begin{bmatrix} x & y & z \end{bmatrix}^T$ will be rewritten as $x = \begin{bmatrix} x & y & z & \theta_x & \theta_y \end{bmatrix}^T$. Because of the structure limitation, it cannot rotate on the $z$-axis, so $\theta_z$ is removed. The tangent vector $t = \begin{bmatrix} t_x & t_y & t_z \end{bmatrix}$ that can be defined by $\begin{bmatrix} \alpha_{13} & \alpha_{23} & \alpha_{33} \end{bmatrix}$ from Equation (16), and $\theta_t$ can be defined as Equation (18), with its differentiation version as Equation (19).

$$\theta_t = \begin{bmatrix} arctan\frac{t_y}{t_z} \\ arctan\frac{t_x}{t_z} \end{bmatrix} \tag{18}$$

$$\dot{\theta}_t = \begin{bmatrix} \frac{\dot{t}_y t_z - t_y \dot{t}_z}{t_z^2 + t_y^2} \\ \frac{\dot{t}_x t_z - t_x \dot{t}_z}{t_z^2 + t_x^2} \end{bmatrix} \tag{19}$$

The time derivative of the tangent vector t and derived angular Jacobian vector can be written as:

$$\dot{t} = J_t\dot{q} = \begin{bmatrix} \alpha_{13} & \alpha_{23} & \alpha_{33} \end{bmatrix}^T, J_t = \begin{bmatrix} J_{t1} & J_{t2} & J_{t3} \end{bmatrix}^T \tag{20}$$

$$J_\theta = \begin{bmatrix} J_{\theta_1} \\ J_{\theta_2} \end{bmatrix} = \begin{bmatrix} \frac{1}{t_z^2 + t_y^2}(t_z J_{t2} - t_y J_{t3}) \\ \frac{1}{t_z^2 + t_x^2}(t_z J_{t1} - t_x J_{t3}) \end{bmatrix} \tag{21}$$

The velocity kinematics is rewritten as Equation (22), and the Jacobian matrix can be factored out from it

$$v_{jacobian} = \begin{bmatrix} v_x & v_y & v_z & \omega_x & \omega_y \end{bmatrix}^T = J\dot{q} = \begin{bmatrix} \alpha_{14} & \alpha_{24} & \alpha_{34} & J_{\theta_1\dot{q}} & J_{\theta_2\dot{q}} \end{bmatrix} \tag{22}$$

The Jacobian matrix combined with forward kinematics can control the posture of the manipulator through the velocity and direction of the end. When W is the identity matrix, the solution obtained by $J(q)^+\dot{x}$ is the least square solution, with $J(q)^+$ is the pseudoinverse of the Jacobian matrix and $I - J(q)^+J(q)$ is the zero-space projection matrix:

$$\dot{q} = J(q)^+ \dot{x} + \{I - J(q)^+J(q)\}\varepsilon \tag{23}$$

$$J^+ = W^{-1}J'\left(JW^{-1}J'\right)^{-1} \tag{24}$$

Next, the arc parameters of each segment are calculated through Equation (23) and the end-effector velocity. Finally, the length of the tension cables of each segment can be calculated through Equations (7)–(10). Then, control of the manipulator posture is achieved.

Finally, the proposed continuum robot's working space and limitations are elaborated, as shown in Table 2 and Figure 9. The manipulator's current end-effector position can be obtained by substituting arc parameters $(\varphi, \kappa, l)$ into the forward kinematics. Then the operating range of the flexible manipulator is in an ideal situation. However, the actual length of the compression springs is limited by $\pm 20$ mm, so after the restriction is substituted in and recalculated, the actual working range of the arm is spherical.

**Table 2.** Limitations of the arc parameters.

| $\theta_1, \theta_2, \theta_3$ | $\varphi_1, \varphi_2, \varphi_3$ | $\kappa_1, \kappa_2, \kappa_3$ |
|---|---|---|
| $0^o \sim 90^o$ | $0^o \sim 360^o$ | $\in R^+$ |

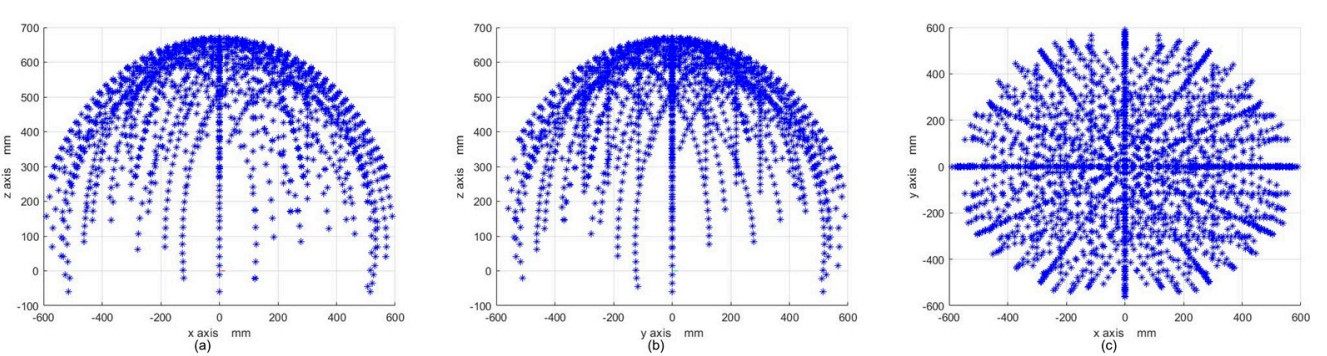

**Figure 9.** Working space analysis of the proposed continuum robot: (**a**) *x*-axis, (**b**) *y*-axis, (**c**) *z*-axis.

### 2.3. Implementation of the Image Servo Tracking Systems

In image-based servo tracking, it is necessary to convert the velocities of the camera and the feature point. The transformation matrix of this conversion is called an Image Jacobian and represents the velocity conversion relation between the feature point and the camera. The camera's location defines the camera's location coordinates, the feature points are the pixel coordinates in the image plane, and the conversion of the two points is considered the projection model of the pinhole camera. In the perspective projection model, the relation between the coordinates of the feature point $(x_f, y_f, z_f)$ in the working space and the projection point $(u, v)$ on the image plane can be written as:

$$\begin{bmatrix} u & v \end{bmatrix}^T = \frac{\lambda}{z_f} \begin{bmatrix} x_f & y_f \end{bmatrix}^T; \quad \lambda \text{ is the focal length} \tag{25}$$

If the camera is placed in a static environment and the camera moves at a translation velocity $V = \begin{bmatrix} V_x & V_y & V_z \end{bmatrix}^T$ and a rotation velocity $\omega = \begin{bmatrix} \omega_x & \omega_y & \omega_z \end{bmatrix}^T$, then the velocity of the feature point $P$ in the camera coordinates can be expressed as:

$$\frac{dP}{dt} = -\omega \times P - V \tag{26}$$

$$\begin{cases} \dot{x}_f = -z_f \omega_y + z_f v \omega_z / \lambda - V_x \\ \dot{y}_f = -z_f u \omega_z / \lambda + z_f \omega_x - V_y \\ \dot{z}_f = -z_f v \omega_x / \lambda - z_f u \omega_y / \lambda - V_z \end{cases} \tag{27}$$

Finally, after differentiating Equation (25) to time and substituting Equation (27), Equation (28) can be factored out where $J_{image}$ is the Image Jacobian, and $V_{camera}$ is the camera's velocity in the camera coordinate system:

$$\begin{bmatrix} \dot{u} \\ \dot{v} \end{bmatrix} = J_{image} V_{camera} = \begin{bmatrix} -\frac{\lambda}{z} & 0 & \frac{u}{z} & \frac{uv}{\lambda} & -\frac{\lambda^2+u^2}{\lambda} & v \\ 0 & -\frac{\lambda}{z} & \frac{v}{z} & \frac{\lambda^2+v^2}{\lambda} & -\frac{uv}{\lambda} & -u \end{bmatrix} \begin{bmatrix} v_x \\ v_y \\ v_z \\ \omega_x \\ \omega_y \\ \omega_z \end{bmatrix} \tag{28}$$

where $(u, v)$ is in length units, but the image obtained in the camera is a series of pixels, so the current $(u, v)$ unit needs to be converted into a pixel unit:

$$\begin{bmatrix} u & v \end{bmatrix}^T = \frac{\lambda}{z} \begin{bmatrix} \frac{x}{S_x} & \frac{y}{S_y} \end{bmatrix}^T; \lambda_x = \frac{\lambda}{S_x}, \lambda_y = \frac{\lambda}{S_y} \tag{29}$$

where $S_x$, $S_y$ and $\lambda$ are the scale and focal length camera intrinsic parameters. Therefore, Equation (28) can be converted as:

$$J_{image} = \begin{bmatrix} -\frac{\lambda_x}{z} & 0 & \frac{u}{z} & \frac{uv}{\lambda_y} & -\frac{\lambda_x{}^2+u^2}{\lambda_x} & \frac{\lambda_x}{\lambda_y}v \\ 0 & -\frac{\lambda_y}{z} & \frac{v}{z} & \frac{\lambda_y{}^2+v^2}{\lambda_y} & -\frac{uv}{\lambda_x} & -\frac{\lambda_y}{\lambda_x}u \end{bmatrix} \tag{30}$$

2.3.1. Eye-To-Hand (ETH) Visual Servoing Configuration

In the ETH configuration, the camera is in a fixed position, so the depth is fixed in the calculation. The camera is obtained as the pixel coordinate system in the image plane, and the manipulator is the robot coordinate system, so it is necessary to multiply the transformation matrix to convert the pixel coordinate system to the spatial coordinate system. The relation can then be formulated as follows:

$$\begin{bmatrix} x_{robot} \\ y_{robot} \end{bmatrix} = [R] \begin{bmatrix} u \\ v \end{bmatrix} + [T] \tag{31}$$

Through the conversion relation, the points in the image plane can be transferred to the robot coordinate system, and through kinematics Equations (14)–(24), the manipulator can perform tasks.

2.3.2. Eye-In-Hand (EIH) Visual Servoing Configuration

The velocity of the camera in the camera coordinate system $v_{camera}$ and the velocity of the camera in the robot coordinate system $v_{jacobian}$ are in different references. Therefore, to combine the two values, a coordinate conversion is required. Thus, Hutchinson et al. [37] proposed the conversion as follows:

$$v_{jacobian} = \begin{bmatrix} R_m v_{tcamera} - R_m v_{\omega camera} \times r_m \\ R_m v_{\omega camera} \end{bmatrix} = \begin{bmatrix} R_m v_{tcamera} + r_m \times R_m v_{\omega camera} \\ R_m v_{\omega camera} \end{bmatrix} = \begin{bmatrix} R_m & sk(r_m)R_m \\ 0 & R_m \end{bmatrix} \begin{bmatrix} v_{tcamera} \\ v_{\omega camera} \end{bmatrix} \tag{32}$$

$$R_m = \begin{bmatrix} T_{m11} & T_{m12} & T_{m13} \\ T_{m21} & T_{m22} & T_{m23} \\ T_{m31} & T_{m32} & T_{m33} \end{bmatrix}, r_m = \begin{bmatrix} T_{m14} \\ T_{m24} \\ T_{m34} \end{bmatrix} \tag{33}$$

$$v_{jacobian} = \begin{bmatrix} R_m & sk(r_m)R_m \\ 0 & R_m \end{bmatrix} V_{camera} \tag{34}$$

Because the camera is installed at the tail of the manipulator, the rotation vector $R_m$ and the translation vector $r_m$ of the camera coordinate system's origin in the robot coordinate system can be obtained as Equation (33). After the calculation, the conversion relationship between the velocity in the camera coordinate system and the camera's velocity in the robot coordinate system can be obtained as in Equation (34).

With the dramatic development of deep learning, deep learning-based computer vision techniques are emerging in various engineering fields. For object detection tasks, the YOLO algorithm [38] and its variants [39,40] provide a reliable detection tool to detect objects in real-time. Furthermore, by integrating with the robot system, some deep vision-based methods were proposed to do the picking task with high accuracy [41,42]. In this study, for plugging and unplugging tasks, we applied the well-known object detection open-source framework, YOLOv4 [40], to detect the plugging area before extracting the target tracking point for action accomplishment.

## 3. Experimental Results

### 3.1. CKM-Based Continuum Robot Implementation

In the proposed three-segment CKM continuum robot, the three CKMs are independent, each controlled separately by three independently actuated tension cables. Therefore, the first CKM circular plates need to pass nine steel cables, the second CKM has six cables, and the third CKM has three cables. In each CKM, non-control tension cables are covered with a tension spring layer, allowing the non-control tension cables to change their length without affecting the posture of the current CKM.

When a tension cable is pulled to the tension-cable control module, it needs to bend because of the different directions, so using a brake housing helps to change direction. Controlling the cable through the brake housing can change the direction but will not affect the force of the cable. Due to the proposed continuum manipulator with three CKMs and nine tension cables, control requires a total of nine sliding tables. Therefore, the nine linear sliding tables are integrated into a rectangular tension-cable control module to facilitate construction. To reduce the total manipulator height, the tension-cable control module is localized horizontally in $3 \times 3$ square layers with three sliding tables in each layer controlling a CKM. After all the control cables pass through the outer brake housings fixed on the aluminum plate, they will be fixed on the slider. Finally, each CKM can be controlled by a group of sliders. The break housings and the sliding tables are illustrated in Figure 10.

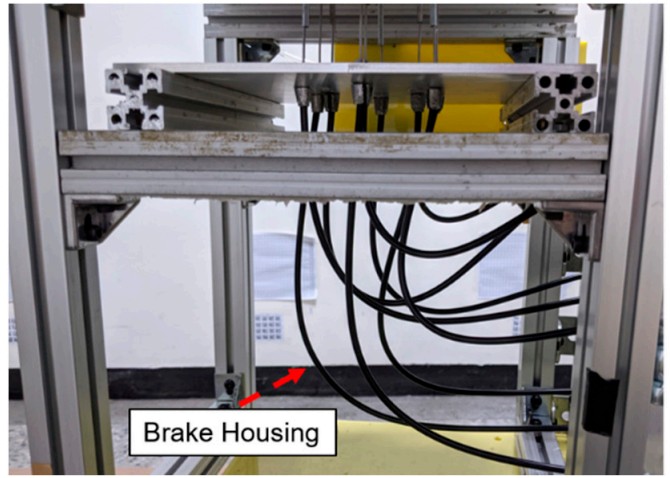 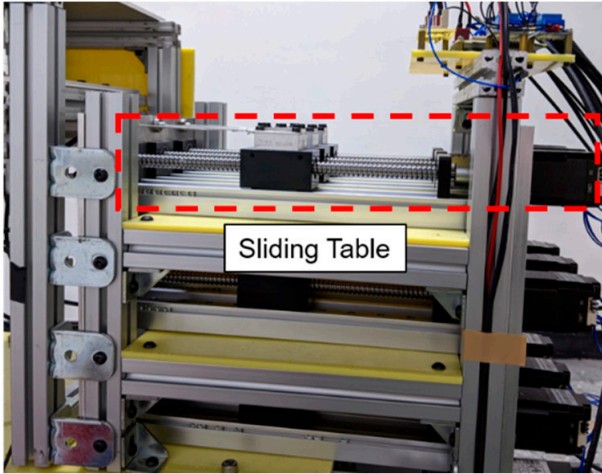

**Figure 10.** The brake housings (**left**) and a single sliding table (**right**).

### 3.2. Kinematic Simulations

Simulation of each kinematics is performed in MATLAB to verify the correctness of the kinematics and posture of the manipulator. First, the arc parameters $(\kappa, \varphi, l)$ are substituted in the kinematics of a single CKM, $\kappa$ is set to 0.003, and $l$ is set to 170. Then, as shown in Figure 11, the simulation of the rotation angle of a single CKM is achieved. Finally, the arc parameters $(\varphi, \kappa, l)$ of each segment are substituted in the kinematics of the flexible manipulator.

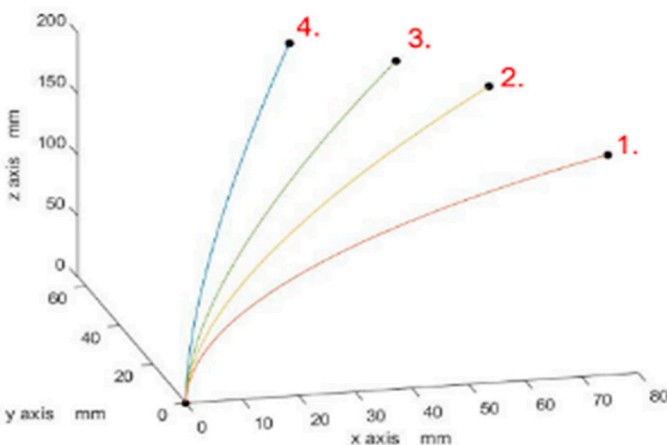

**Figure 11.** Simulation of the rotation angle $\varphi$ of single CKM kinematic: **1.** $\varphi = 0$; **2.** $\varphi = \pi/2$; **3.** $\varphi = \pi/4$; **4.** $\varphi = \pi/3$.

In Figure 12a, substituting the same rotation angle $\varphi$ in the three segments, the rotation angle $\varphi$ is $(0 - 2\pi$, scale of $\pi/4)$, and the curvatures of the three segments $\kappa$ are set as 0.003, 0.002, and 0.001, respectively. In Figure 12b, the rotation angle $\varphi$ is substituted in the first and third segments, the rotation angle $\varphi + \pi$ is substituted in the second segment, and the rotation angle $\varphi$ is $(0 - 2\pi$, scale of $\pi/4)$, where the curvatures $\kappa$ of three segments are 0.003, 0.002, and 0.001, respectively.

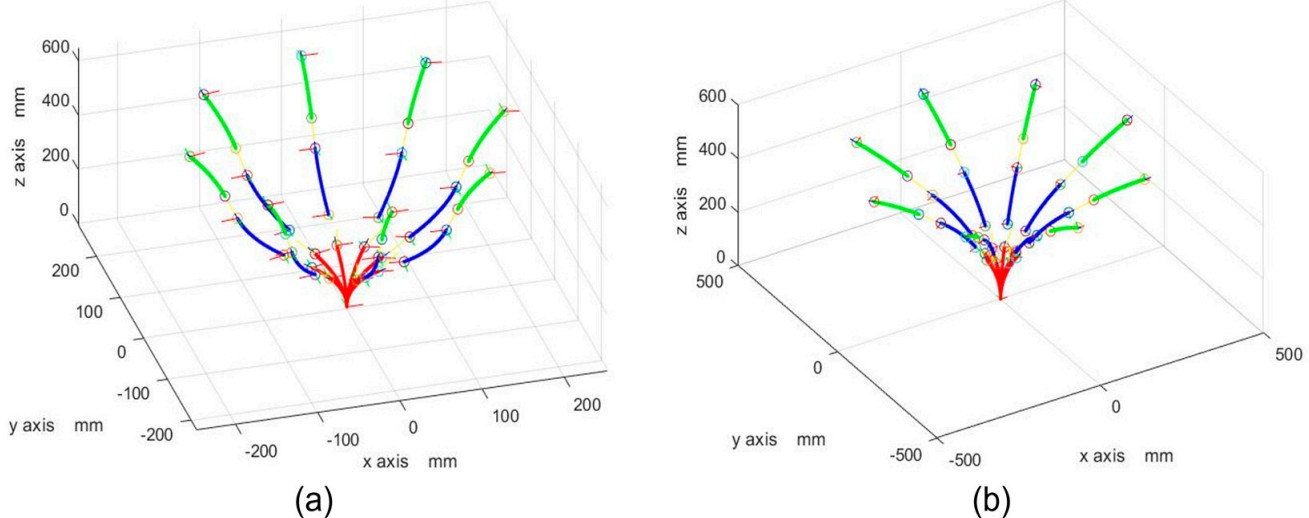

**Figure 12.** Kinematic simulations of the flexible manipulator: (**a**) same rotation angle $\varphi$ in three segments, (**b**) rotation angle $\varphi$ in the first segment (red) and third segment (green), rotation angle $\varphi + \pi$ in the second segment (blue).

Next, the change in the curvature $\kappa$ is simulated. The rotation angles $\varphi$ for the three segments are fixed, and the order is $\pi/3$, $4\pi/3$, and $\pi/3$. Figure 13a shows that the arc curvature $\kappa$ of the second segment changes from 0.002 to 0.007, each time increasing by 0.001, and the curvature $\kappa$ of the first and third segments are fixed sequentially to 0.003 and 0.001. Figure 13b shows that the arc curvature $\kappa$ of the third segment changes from 0.001 to 0.021, each time increasing by 0.005, and the curvature $\kappa$ of the first and third segments are fixed sequentially to 0.003 and 0.002, respectively.

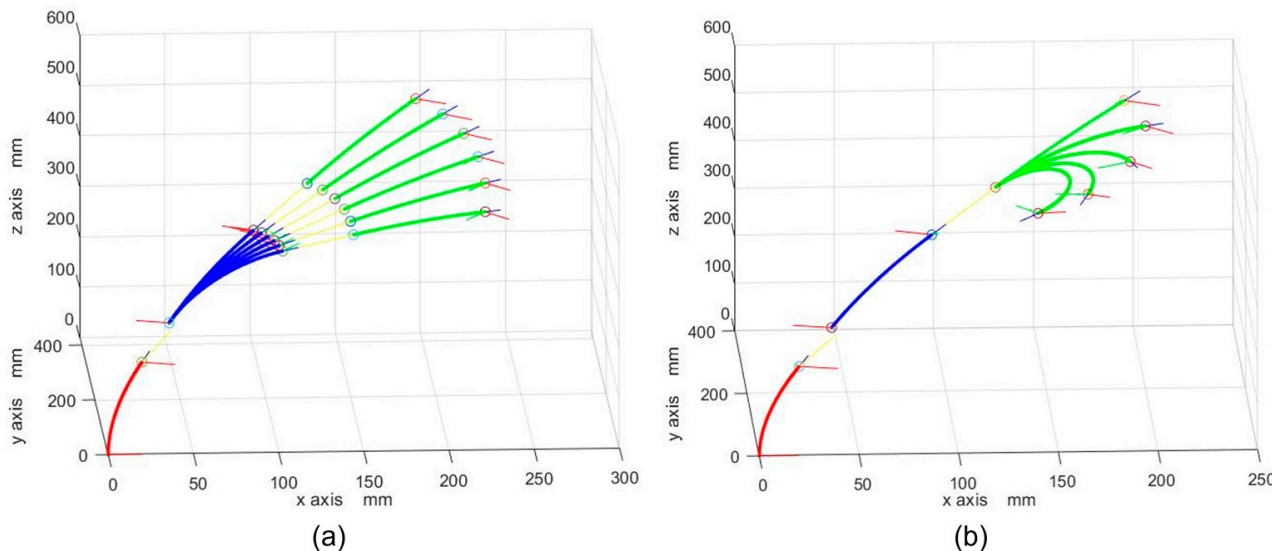

(a)

(b)

**Figure 13.** Kinematic simulation of the flexible manipulator: (**a**) change in the arc curvature $\kappa$ of the second segment, (**b**) change in the arc curvature $\kappa$ of the third segment.

The last is the simulation of velocity kinematics, as shown in Figure 14. Given an initial posture of the flexible manipulator, the first segment arc parameter $(\kappa, \varphi, l)$ is (0.003, 0, 170), the second segment arc parameter $(\kappa, \varphi, l)$ is (0.002, $\pi/2$, 170), and the third segment arc parameter $(\kappa, \varphi, l)$ is (0.001, $\pi/2$, 170). The kinematics are simulated by substituting different end velocities and directions in the kinematics. Finally, the kinematics have been verified for feasibility.

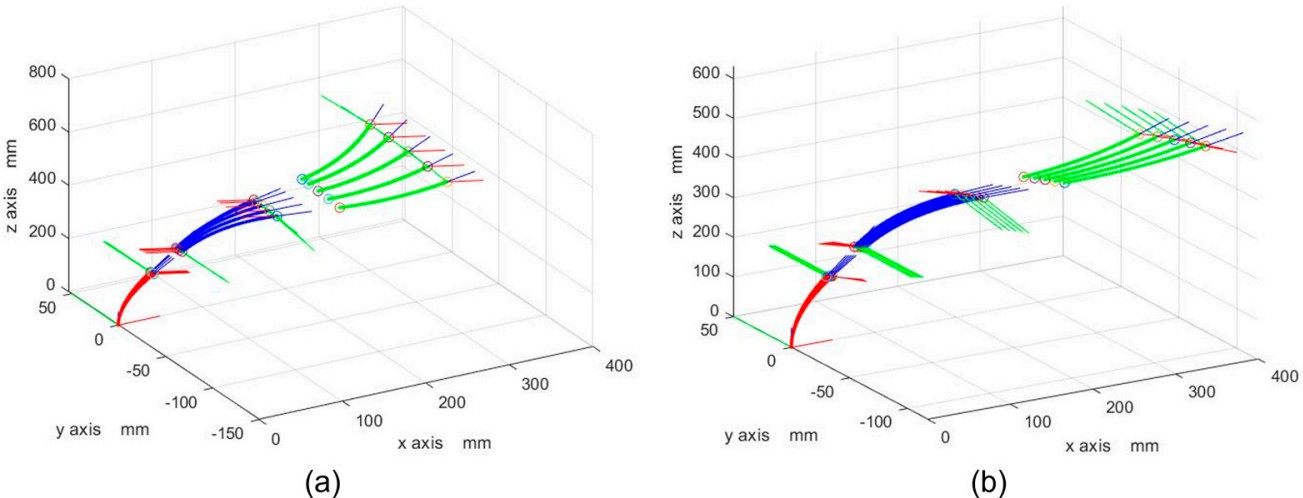

(a)

(b)

**Figure 14.** Velocity kinematic simulation of the flexible manipulator: (**a**) change in the end velocity [0 20 0 0 0 0] (**b**) change in the end velocity [20 0 0 0 0 0].

The posture of the arm movement over 4 s is simulated, as shown in Figure 14. The endpoint records of the two simulation results of each second are shown in Table 3. The end velocity vector (0, 20, 0, 0, 0, 0), which means that it moves 20 mm per second along the positive *y*-axis during the end velocity vector ( 20, 0, 0, 0, 0, 0), which means that it moves 20 mm per second in the positive *x*-axis. Table 3 shows that the error was within 10 mm, and the maximum error in the *z*-axis was within 50 mm. The feasibility of kinematics can be verified through the above simulations, whether using single CKM kinematics, flexible manipulator kinematics, or velocity kinematics.

**Table 3.** Velocity kinematics simulations-Endpoint error of the manipulator posture.

| Speed Variation | (0, 20, 0, 0, 0, 0) | | | | | |
|---|---|---|---|---|---|---|
| | **X** | **Y** | **Z** | **X_error** | **Y_error** | **Z_error** |
| Start point | 279.6 | −101.5 | 586.8 | - | - | - |
| 1 | 281.6 | −79.38 | 591.0 | −1.8 | −2.12 | −4.2 |
| 2 | 282.7 | −57.43 | 594.4 | −3.1 | −4.07 | −7.6 |
| 3 | 283.5 | −36.03 | 596.8 | −3.9 | −5.47 | −10.0 |
| 4 | 283.6 | −16.42 | 598.4 | −4.0 | −5.08 | −11.5 |
| **Speed variation** | **(20, 0, 0, 0, 0, 0)** | | | | | |
| | **X** | **Y** | **Z** | **X_error** | **Y_error** | **Z_error** |
| Start point | 279.6 | −101.5 | 586.8 | - | - | - |
| 1 | 298.4 | −103.5 | 575.5 | 1.2 | −2.0 | 11.3 |
| 2 | 316.7 | −105.3 | 563.3 | 2.9 | −3.8 | 23.5 |
| 3 | 334.3 | −106.7 | 551.2 | 5.3 | −1.4 | 35.6 |
| 4 | 351.2 | −107.7 | 538.4 | 8.4 | −2.4 | 48.4 |

*3.3. Heart Shape Trajectory Tracking*

In this experiment, the trajectory tracking task of eye-to-hand (ETH) is used to verify the effectiveness of the Jacobian control method. The experimental environment is shown in Figure 15. The camera position was fixed in the experiment, and the distance from the laser point imaging screen was 0.23 m. The exact size heart shape with different numbers of tracking points is generated in the image plane. Finally, the manipulator achieves trajectory tracking through velocity kinematics. The reference trajectory is the heart shape trajectory point defined in the image plane, with the generated heart shape trajectory point calculated using the following formula:

$$
\begin{cases}
y(x) = 60 - \sqrt{\left(1600 - (x - 40)^2\right)} \\
y(x) = 60 - 20(arcos(1 - (x/40)) - \pi)
\end{cases}
\tag{35}
$$

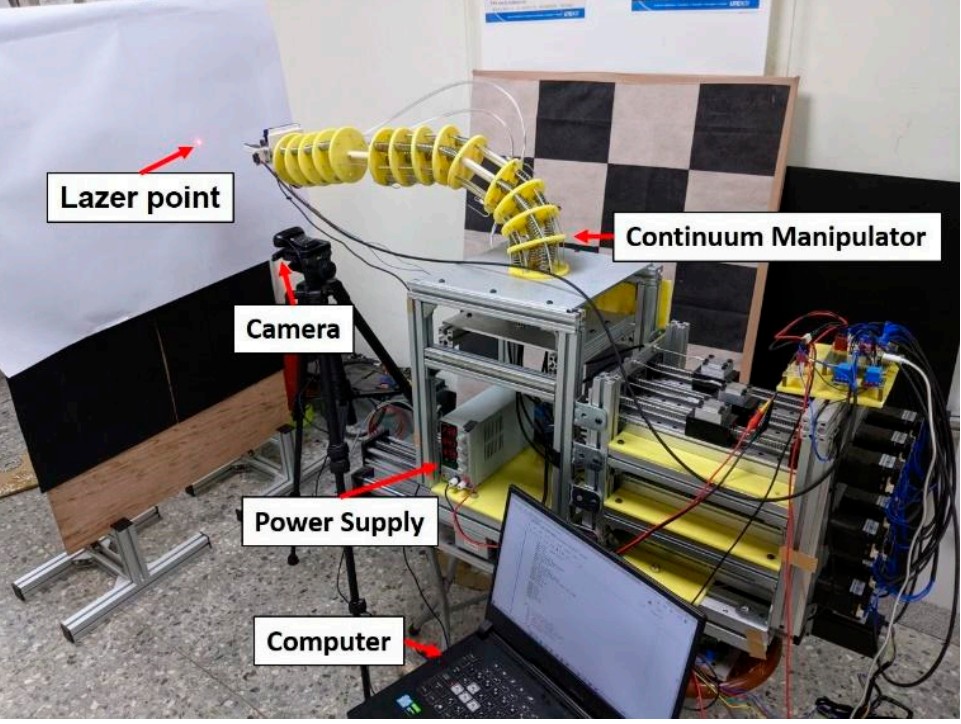

**Figure 15.** The environment of the trajectory tracking.

According to the number of different heart shape trajectory points, five experiments were performed, and the results were averaged to verify the stability and error of the tracking. As a result, the tracking trajectory is shown in Figure 16. The error is the distance between the current laser and target points.

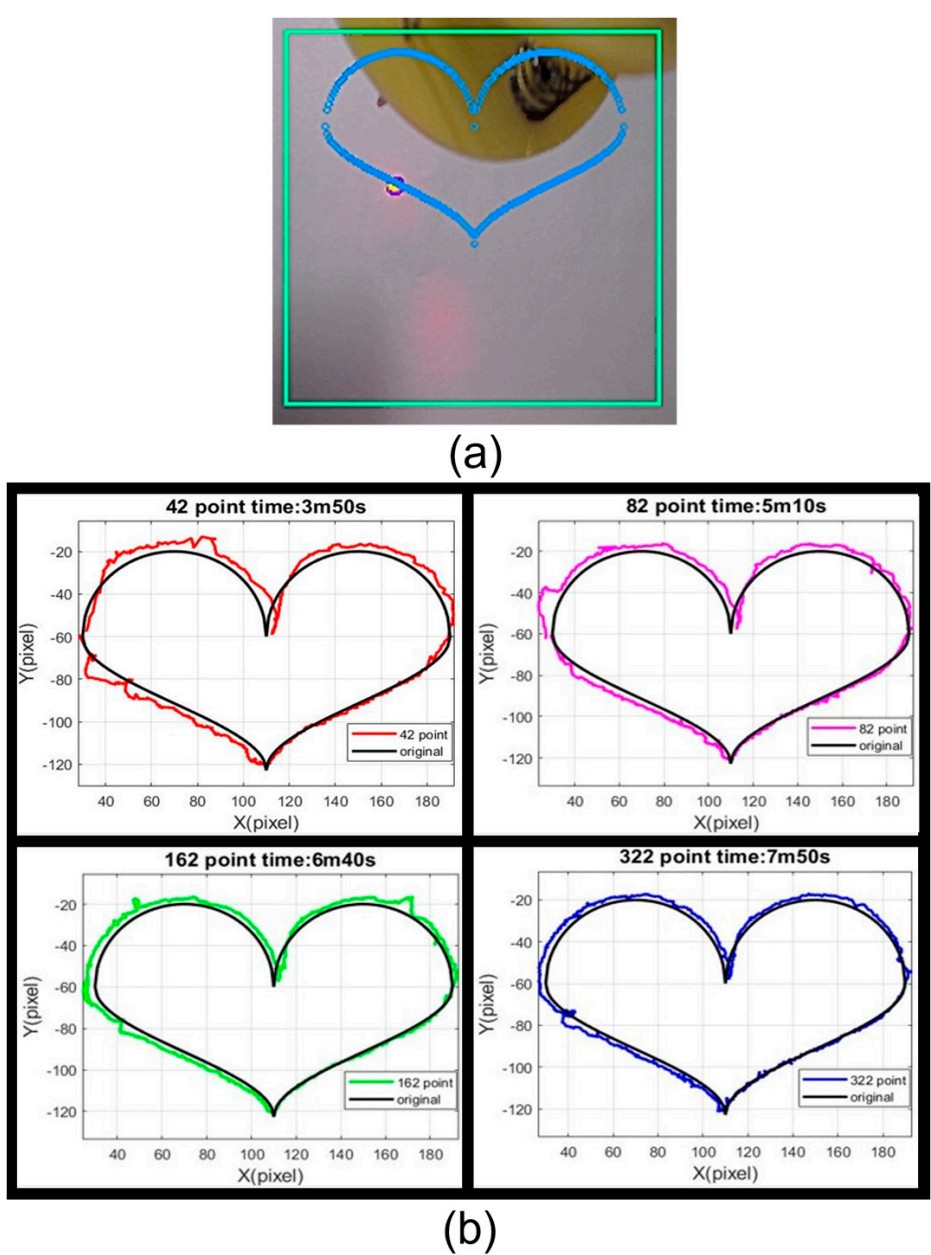

**Figure 16.** Heart shape trajectory tracking (**a**) Image obtained during trajectory tracking (**b**) Tracking trajectory with different numbers of tracking points.

The experimental results are shown in Table 4. The experiment shows that the proposed control method can achieve accurate path tracking.

### 3.4. Image-based Servo Tracking Experiment

The effectiveness and feasibility of image-based servo tracking are verified by the plugging/unplugging task of eye-in-hand (EIH), with a stereo camera and pneumatic grippers installed at the manipulator's end. The experimental environment and the processes of autonomous "plug-in" and "unplug" of electric power sockets are shown in Figures 17 and 18, respectively.

**Table 4.** Velocity kinematics simulations-Endpoint error of the manipulator posture.

| Number of Points | Test | Max Error (pixel) | | Min Error (pixel) | | Average Error (pixel) | | RMSE |
|---|---|---|---|---|---|---|---|---|
| | | X | Y | X | Y | X | Y | |
| 42 | 1 | 14.43 | 27.00 | 0.10 | 0.04 | 3.70 | 5.21 | 7.80 |
| | 2 | 15.23 | 32.30 | 0.02 | 0.41 | 3.89 | 5.34 | 7.67 |
| | 3 | 16.34 | 35.05 | 0.04 | 0.10 | 3.64 | 5.17 | 7.63 |
| | 4 | 15.69 | 28.31 | 0.03 | 0.39 | 3.77 | 5.26 | 7.74 |
| | 5 | 16.89 | 33.41 | 0.21 | 0.19 | 3.98 | 5.31 | 7.87 |
| | Average | 15.72 | 31.21 | 0.08 | 0.22 | 3.80 | 5.26 | 7.74 |
| 82 | 1 | 40.00 | 19.24 | 0.00 | 0.33 | 2.82 | 4.08 | 6.05 |
| | 2 | 38.20 | 20.45 | 0.00 | 0.33 | 2.79 | 3.99 | 6.08 |
| | 3 | 42.32 | 18.78 | 0.08 | 0.07 | 2.67 | 4.12 | 6.13 |
| | 4 | 39.64 | 19.67 | 0.02 | 0.09 | 2.37 | 4.05 | 6.09 |
| | 5 | 37.25 | 20.26 | 0.07 | 0.05 | 2.88 | 3.98 | 6.03 |
| | Average | 39.50 | 19.68 | 0.03 | 0.17 | 2.71 | 4.04 | 6.08 |
| 162 | 1 | 36.00 | 14.42 | 0.00 | 0.06 | 1.81 | 3.09 | 4.26 |
| | 2 | 37.23 | 14.36 | 0.02 | 0.02 | 1.86 | 3.07 | 4.32 |
| | 3 | 34.05 | 13.96 | 0.00 | 0.03 | 1.82 | 3.08 | 4.23 |
| | 4 | 32.35 | 14.85 | 0.00 | 0.04 | 1.83 | 3.02 | 4.36 |
| | 5 | 35.87 | 13.56 | 0.02 | 0.01 | 1.79 | 3.08 | 4.33 |
| | Average | 35.10 | 14.23 | 0.01 | 0.03 | 1.82 | 3.07 | 4.30 |
| 322 | 1 | 6.02 | 11.09 | 0.02 | 0.00 | 1.79 | 3.10 | 3.97 |
| | 2 | 6.04 | 11.05 | 0.01 | 0.02 | 1.80 | 2.97 | 4.07 |
| | 3 | 5.98 | 11.09 | 0.00 | 0.03 | 1.73 | 2.99 | 4.05 |
| | 4 | 6.06 | 11.12 | 0.00 | 0.01 | 1.75 | 3.12 | 4.01 |
| | 5 | 6.03 | 10.99 | 0.00 | 0.01 | 1.76 | 3.08 | 4.03 |
| | Average | 6.03 | 11.07 | 0.01 | 0.02 | 1.77 | 3.05 | 4.03 |

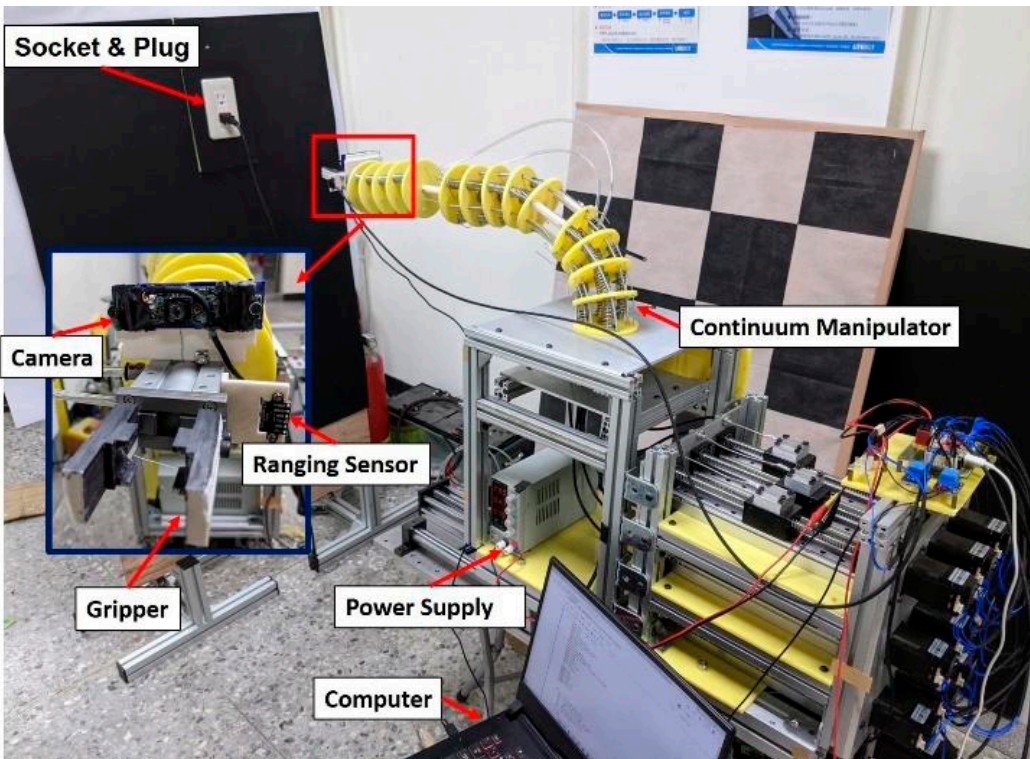

**Figure 17.** Environment for the plugging/unplugging experiment.

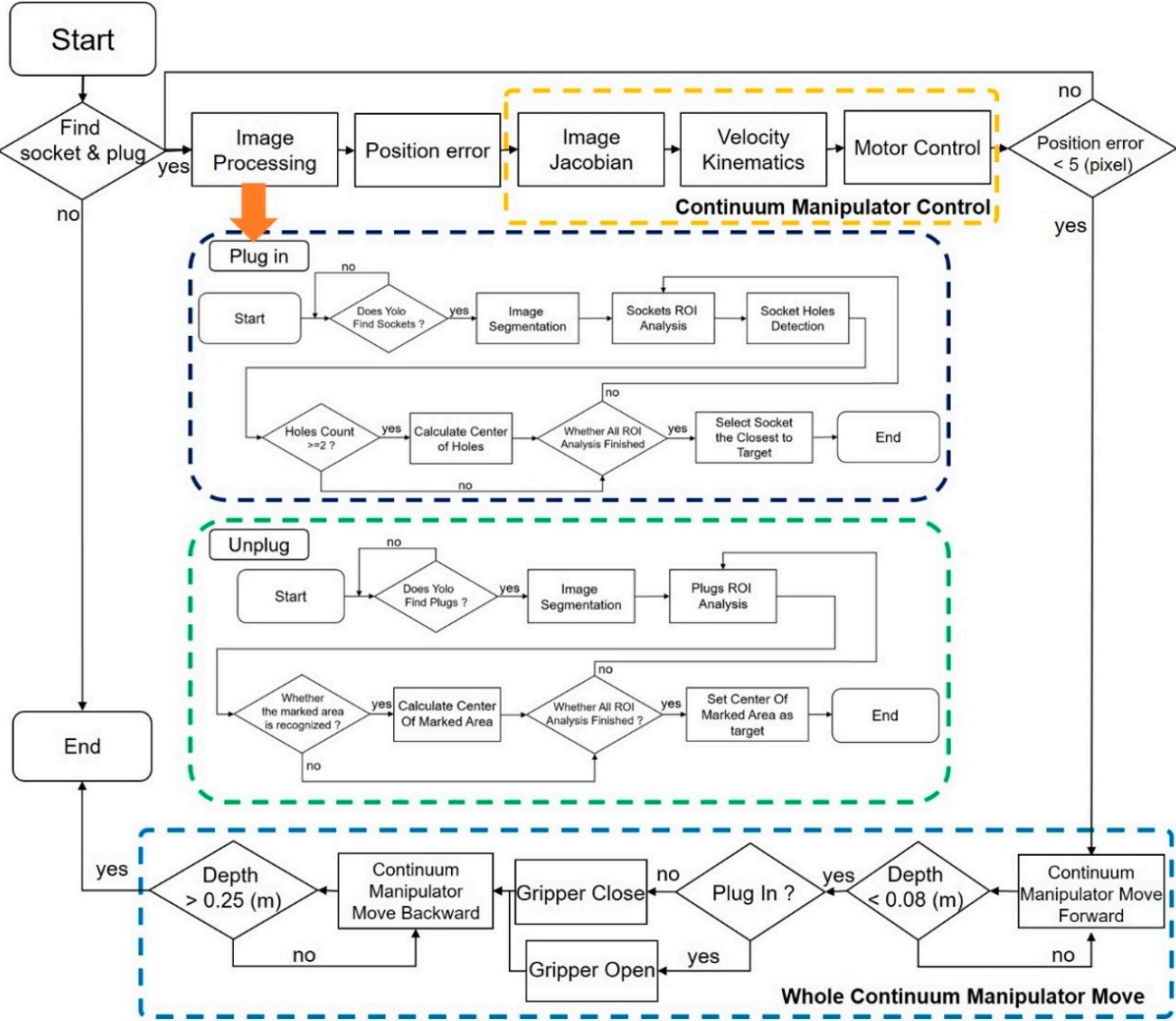

**Figure 18.** The algorithm flowchart of the Plugging/Unplugging task.

The experiment has two main parts: plugging and pulling the plug. In both experiments, the socket and plug specifications are in Figure 19. A widely used object detection open-source framework, YOLOv4 [39], was used to identify the target area first, and then the target tracking point was analyzed through image processing. Figure 20 shows the results of the detected electric socket and plug.

After the target tracking point is obtained, the error can be calculated between the current position and the tracking point. The arc parameters $(\varphi, \kappa, l)$ of each segment of the manipulator can be obtained by substituting the error into the Image Jacobian and velocity kinematics. Then, the manipulator can perform the task of plugging/unplugging through arc parameters $(\varphi, \kappa, l)$.

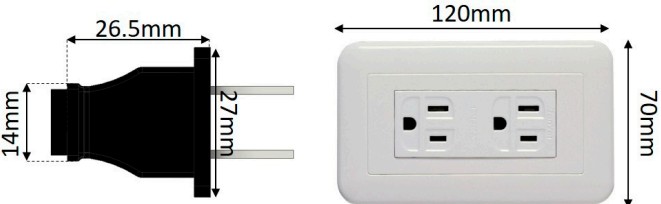

**Figure 19.** Specifications of the socket and plug.

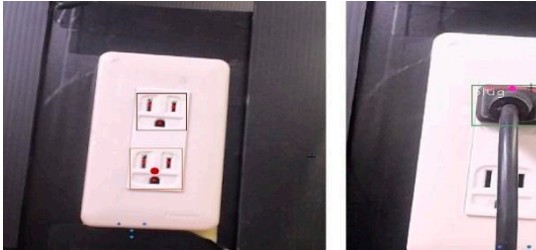
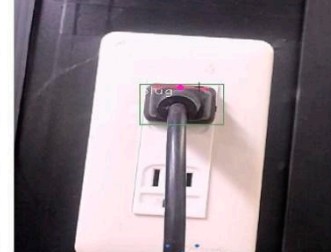

**Figure 20.** Object detection results using YOLO for the electric power socket (left-hand-side) and the electric power plug (right-hand-side). It is noted that the detected object areas are indicated as rectangle boxes.

Finally, Figures 21 and 22 show the results of the plugging and unplugging experiments, respectively. Both parts of the experiment (plugging-in and plugging-out) were performed five times, and the experimental results are shown in Table 5. We set the start points for each experimental trial as the final points, around 50 cm in front of the target. For five trials of unplugging, the success rate is 100%. However, the plugging-in experiments show a success rate of only 60%. The visual servo system could achieve lows of $58 \pm 2.12$ s and $83 \pm 6.87$ s in processing time at the 90% confidence level in the unplugging and plugging-in tasks, respectively. The main reason is that the power plugging-in into the socket needs a higher accuracy because of the small holes in the socket. Therefore, the control accuracy and rigidity of the continuum manipulator should be further improved in the future.

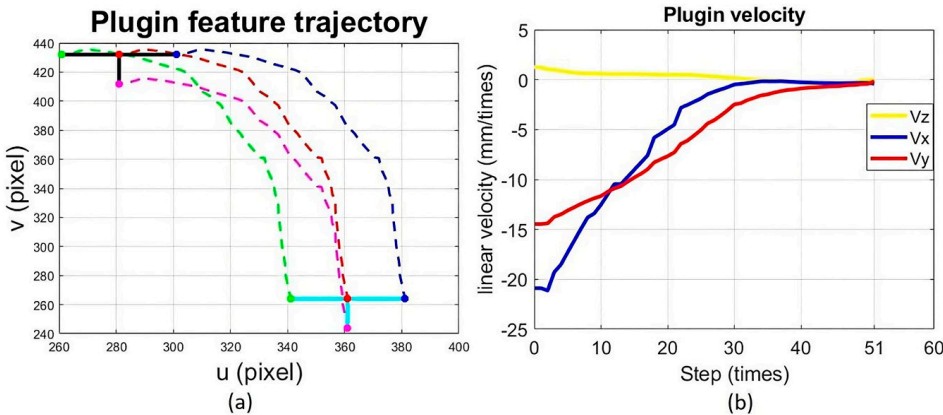

**Figure 21.** Plugging-in experiment: (**a**) trajectory of the feature points, (**b**) end-effector linear velocity $v_x$, $v_y$ and $v_z$.

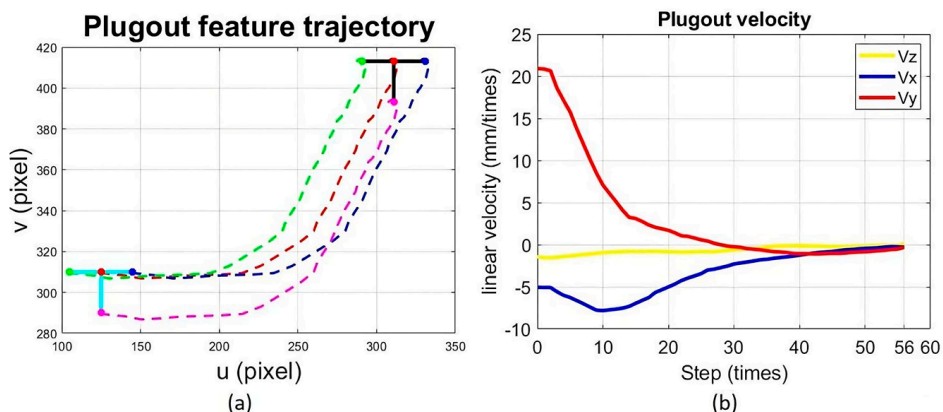

**Figure 22.** Unplugging experiment: (**a**) trajectory of the feature points, (**b**) end-effector linear velocity $v_x$, $v_y$ and $v_z$.

**Table 5.** Plugging/unplugging experimental results for each test.

| | Unplugging Task | | Plugging Task | |
|---|---|---|---|---|
| | Time (s) | Mission Completion | Time (s) | Mission Completion |
| 1 | 61 | Success | 88 | success |
| 2 | 55 | Success | 80 | success |
| 3 | 62 | Success | - | fail |
| 4 | 58 | Success | 75 | success |
| 5 | 57 | Success | - | fail |

## 4. Conclusions

In this paper, a continuum kinematic module (CKM) was proposed. The function and feasibility of the CKM were demonstrated using a continuum manipulator composed of three CKMs. First, the continuum manipulator's kinematics were verified through a kinematic simulation and analysis, and the correctness of the manipulator posture was confirmed through the simulation. Then, an ETH heart shape tracking experiment was established to verify the correctness of the kinematics and the accuracy of control of the continuum manipulator in practical applications. Finally, the accuracy and feasibility of image servo tracking in an EIH plugging-in/unplugging experiment were demonstrated in extensive tolerance conditions. At the same time, the results show that the accuracy needs to be improved in low-tolerance conditions. Furthermore, the wire-driven CKM easily causes posture deviation due to the influence of gravity.

In future work, an inertial measurement unit (IMU) could be installed to correct posture deviation through feedback. To increase the accuracy of the image-based servo tracking, the current IBVS can be changed to PBVS because the IBVS method does not involve manipulator posture considerations. The PBVS will instead estimate the current pose of the target relative to the camera. Because the continuum manipulator has a greater degree of freedom, the wire-driven manipulator is susceptible to the influence of gravity, which causes posture deviation. Therefore, if PBVS-based image-based servo tracking is used, the accuracy of the image-based servo tracking may be improved. In addition, the continuum manipulator is limited in movement. Therefore, it is proposed that the mounting of the robot on an AGV can remove the restriction in the movement of the manipulator, and indoor positioning will allow the manipulator to perform more diversified tasks. Finally, the continuum manipulator can be applied to collaborative or service robots.

**Author Contributions:** Methodology, M.-H.H.; Project administration, C.-H.K.; Software, M.-H.H.; Supervision, C.-H.K.; Validation, M.-H.H.; Visualization, M.-H.H.; Writing—original draft, M.-H.H.; Writing—review & editing, M.-H.H., P.T.-T.N. and D.-D.N. All authors have read and agreed to the published version of the manuscript.

**Funding:** This research received no external funding.

**Institutional Review Board Statement:** Not applicable.

**Informed Consent Statement:** Not applicable.

**Data Availability Statement:** Not applicable.

**Conflicts of Interest:** The authors declare no conflict of interest.

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
