# Peer review of "Image Servo Tracking of a Flexible Manipulator Prototype with Connected Continuum Kinematic Modules"

_actuators, doi:10.3390/act11120360_

Round 1

Reviewer 1 Report

The paper presents a flexible manipulator which is composed of some connected kinematic modules. The operation of the proposed approach is showed with image-based servo tracking tasks. The work is interesting. Notwithstanding that, there is a number of issues that the authors should address to clearly show their contribution and to prove the advantages of the proposal.

First of all, the authors should more clearly describe how their proposal frames in the current state-of-the-art, both considering the literature on continuum robots and also the literature on visual control. To do it, section 2 (related work) should be completed with some additional information to frame the proposal and to clearly present the advantages, disadvantages and the improvements that the authors achieve with respect to recent works in the state-of-the-art.

Also, section 2 contains some sentences that must be clarified. For example, the authors should explain what they mean with these two sentences:

- "and the results showed that 6.5% appeared in the key point positioning error based on the manipulator's length".

- "Their results showed a 25.23% positioning error compared to the original error".

It is not clear which is the process that the authors follow to obtain the data presented in table 3. Please describe. Also, in table 5, in all the experiments: are the initial and final positions the same?

Fig. 9 could be removed, as it contains very well known information about the projection model.

Finally, it is very important that the authors describe which are the practical issues that they have found while implementing the robot and the control. For example, have the authors found any issue due to the fact of using tension cables to control the CKMs? Also, for example, is the visual control very sensible to the parameters obtained after the calibration process? Please develop so that the reader can have a clear idea of the practical implementation issues.

The authors must carefully proofread the paper. There are several issues with the use of English language that make some passages hard to follow.

Author Response

Please see our reply-to-referees files, thanks!

Reviewer 2 Report

Brief summary

This paper presents the design and implementation of a flexible manipulator formed with connected continuum kinematic modules. Its main contribution consists in ease the fabrication of a continuum robot with multiple degrees of freedom.

Broad comments

The document is sometimes not easy to read and follow.

The English needs minor review and spell checking.

The document is well supported with references.

The subject of the paper is interesting and with a great potential of application.

One of the weaknesses of this study is that the plug-in experiments show low accuracy.

Also, authors reference many consecutive figures and/or tables in the text which forces the reader to scroll up and down the document very frequently. This should be corrected.

Specific comments

In line 69 did you mean “trunk” ?

Authors should reference figures in the text before showing them (eg. Figure 1, 3, 4, 9, …). Please correct.

Authors should include a short explanation of each variable after introducing each equation. Please correct.

Authors should discuss the results obtained in Table 5 including the time taken for unplugging and plugging tasks (close or above 1 minute).

Author Response

Please read our reply to referees document. Thank you very much!

Reviewer 3 Report

In this paper, a continuum kinematic module was proposed. The function and feasibility of the CKM were demonstrated using a continuum manipulator composed of three CKMs. First, the continuum manipulator's kinematics were verified through a kinematic simulation and analysis, and the correctness of the manipulator posture was confirmed through the simulation. Then, the ETH love tracking experiment was established to verify the correctness of the kinematics and the accuracy of control of the continuum manipulator in practical applications. Finally, the accuracy and feasibility of image servo tracking in the EIH plugging/unplugging experiment were demonstrated in large tolerance conditions, In summary, the research is interesting and provides valuable results, but the current document has several weaknesses that must be strengthened in order to obtain a documentary result that is equal to the value of the publication.

General considerations:

(1) Concerning the presentation of the contents, the document is acceptable. Nonetheless, it is recommended that authors develop proofreading to avoid common mistakes such as incorrect expressions and incorrect use of punctuation rules(e.g. in line 150: “Figure 1.,”, in line 159: “Figure. 2.”, or in line 361 03.1 and 2,9).

(2) The document contains a total of 30 employed references, of which 21 are publications produced in the last 5 years (70%), 5 in the last 5-10 years (17%), 4 than 10 years old (13%) , implying a total percentage of 87 % recent references. In this way, the total number of references is insufficient.

Title, Abstract and Keywords:

(3) The abstract is complete and well-structured and explains the contents of the document very well. Nonetheless, the abstract mainly introduces the composition of flexible manipulator, and the means and methods used in the research work should be increased, and the results and important conclusions are drawn. The part relating to the results could provide numerical indicators obtained in the research.

Chapter 1: Introduction

(4) The first paragraph introducing the research topic gives a too simple, and even incomplete, view of the problems related to your topic and should be revised and completed with citations to state-of-art references (Yunchao Tang; Zhaofeng Huang; Zheng Chen; Mingyou Chen; Hao Zhou; Hexin Zhang; Junbo Sun. Novel visual crack width measurement based on backbone double-scale features for improved detection automation, Engineering Strucutres 2023, 274: 115158. https://doi.org/10.1016/j.engstruct.2022.115158). 

(5) The novelty of the study is not apparent enough. In the introduction section, please highlight the contribution of your work by placing it in context with the work that has done previously in the same domain.

(6) On a general level, the study of the proposed design techniques is reasonable, and the explanation of the objectives of the work may be valid. However, the limitations of your work are not rigorously assumed and justified.

Chapter 3: Proposed Method

(7) There appears to be no indication of the computational tools and software resources used to carry out the methods presented.These issues could be presented in a more orderly and clear manner.

(8) Vision technology integrated with deep learning is emerging these years in various engineering fields. For object detection, please refer to A study on long–Close distance coordination control strategy for litchi picking; Fruit detection and positioning technology for a Camellia oleifera C. Abel orchard based on improved YOLOv4-tiny model and binocular stereo vision.

Chapter 4: Experimental results

(9) In Table 3, some numerical calculations of the endpoint error of the  manipulator posture are incorrect.

Chapter 5: Conclusions

(10) It should mention the scope for further research as well as the application of the study.

Author Response

(The authors gave the same response as above.)

Round 2

Reviewer 1 Report

The paper has somewhat improved with the revision. Notwithstanding that, I still have some concerns about the manuscript:

- The authors have completely rewritten and completed the introduction. It is nice that they now include more complete information about the state-of-the-art. However, in my opinion, the current introduction is excessively long and it should be divided into two sections: (a) a briefer introduction that presents the gist of the problem and (b) a complete description of the state of the art.

- I still have concerns about the contributions of the paper. As I stated in the previous review, I think that the authors should more clearly frame their work in the current state-of-the-art, more clearly describing the contributions, advantages and disadvantages of their proposal. The authors have included four bullet points at the end of the introduction, but as I see it, most of them cannot be considered as contributions of the paper. For example, about the second bullet point (line 144): which is exactly the contribution in the field of ETH and EIH control? Also, bullet points 3 and 4 (lines 148 and 157) are mere descriptions of the ETH and EIH experiments, but not contributions of the present work.

- Why have some numerical values in table 3 changed with respect to the previous version of the paper?

- About the experiments in section 3.4, I think that the authors should expand the range of experiments (for example, considering a variety of starting and ending points) to more clearly prove the validity of the approach.

- I still think that the paper should be carefully proofread.

Author Response

Please read the reply to referees documents. Thank you very much!

Reviewer 3 Report

Congrats! The authors have successfully addressed all my comments. Therefore, I recommend the publication of this manuscript.

Author Response

(The authors gave the same response as above.)

Round 3

Reviewer 1 Report

The paper has clearly improved with the revisions, and the contributions and limitations of the work are now more clearly presented. In my opinion, the paper can be accepted in current form.